# Individual Scent-Marks of Nest Entrances in the Solitary Bee, *Osmia cornuta* (Hymenoptera: Apoidea)

**DOI:** 10.3390/insects12090843

**Published:** 2021-09-18

**Authors:** Konrad Sebastian Frahnert, Karsten Seidelmann

**Affiliations:** 1Weinberg-Gymnasium Kleinmachnow, Am Weinberg 20, D-14532 Kleinmachnow, Germany; konrad.frahnert@gmx.de; 2Abteilung Tierphysiologie, Institut für Biologie/Zoologie, Martin-Luther-Universität Halle-Wittenberg, Hoher Weg 8, D-06099 Halle, Germany

**Keywords:** nest recognition, Dufour’s gland, cuticular hydrocarbons, scent mark

## Abstract

**Simple Summary:**

*Osmia cornuta* is a solitary mason bee that uses natural pre-existing cavities, such as beetle worm traces or hollow plant stalks, for nest construction. Such nesting opportunities can often be found pretty close to each other, leading to dense aggregations of many nesting bees. Therefore, it is crucial for females returning from collecting flights to localize their own nests’ entrance among many other similar-looking holes. Individual scent marks of entrances were suspected to be used by bees as olfactory cues additional to visual orientation. We used paper sleeves introduced in the nest entrances to identify the substances involved in marking and compared the composition with secretions of the Dufour’s gland and substances present on the body surface. Although the nest marks were found to be ample diverse, tags did not perfectly render individual odor bouquets nor did bees possess private chemicals. Instead, females used an individual mixture of body-derived substances enriched by external components to produce distinguishable tags that alter over time. The smell of the own nest has to be learned continuously by the resident female as a template to identify her own nest at the next arrival from a provision flight.

**Abstract:**

The ability to recognize the own nest is a basic skill in nest constructing solitary bees. *Osmia cornuta* females use a dual mechanism of visual orientation to approach a nest and olfactory verification of the tube when entering it. Occupied tubular cavities were steadily marked by the resident female. Nest marking substances originate from Dufour’s gland and cuticle, enriched by external volatiles. Scent tags were dominated by alkanes and alkenes in a species-specific mixture enriched by small amounts of fatty acid esters, alcohols, and aldehydes. The individual nest tags are sufficiently variable but do not match perfectly with the nesting female. Furthermore, tags are not consistent over time, although females continue in marking. Besides the correct position of the entrance in space, bees have to learn also the bouquet of the used cavity and update their internal template at each visit to recognize their own nest by its actual smell. Due to the dominance of the species-specific hydrocarbon pattern, nest marks may function not only as an occupied sign but may also provide information on the species affiliation and constitution of the nest owner.

## 1. Introduction

Bees are central-place foragers that store the provisions they have collected at flowers in brood cells of a nest. Females are able to carry small loads only that are not sufficient for the development of a single progeny. They have to navigate hundreds of times over several days back and forth between floral resources or sources of nest-building material (wet mud, leaves, plant wool, etc.) and their nest, thereby bridging considerable distances. On their way back, bees use visual cues to locate their nest [1,2,3,4,5,6]. However, if the nesting substrate is favorable or many nesting cavities are abundant, solitary bees may also nest in dense aggregation with many similar-looking nest entrances side by side. In such cases, visual orientation might not be sufficient anymore to locate the own nest. However, bees possess a reinsurance mechanism and are able to identify their own nest by olfactory cues. This nest recognition by olfactory cues was proved for several species of solitary bees by behavioral experiments [5,7,8,9,10,11].

Some bee species that are abundant or of economic importance as commercial pollinators, such as mason bees (*Osmia*) or the alfalfa leaf-cutting bee *Megachile rotundata* (F.), use preexisting cavities, such as beetle burrows in dead wood, holes in other materials, hollow plant stalks, and also accept artificial tubes. In these species, nest recognition and marking behavior has been studied in more detail [12,13]. Based on observations and experiments, sources of volatiles used for marking the nest tube were hypothesized to be Dufour’s gland (an exocrine gland of the abdomen associated with the sting apparatus in female Apocrita) secretions [7,9,14], mandibular gland secretions [10,15,16], or the cuticular hydrocarbons (CHC) of the body surface [17]. By using glass tubes as nesting material, the application of secretions from the abdominal tip was directly observed in *Osmia lignaria* (Say) [13]. Moreover, the chemical compositions of Dufour’s gland secretions and the CHC bouquet have been analyzed for the blue orchard bee, *O. lignaria,* and the alfalfa leaf-cutting bee, *M. rotundata* [12,13,17], in detail. The occurrence of substances from the Dufour’s gland and the cuticle in nest tubes has been demonstrated for both species in these studies. However, the composition of scent marks, their variability, and the similarity between individual females and their marks, which are basic for olfactory nest recognition, has not been addressed yet.

We used a small artificial colony of the horn-faced mason bee *Osmia cornuta* (Latreille) to analyze the chemical basis of nest recognition in this species. In a first step, the visual guided approach to the nest tube and the identification of the own nest entrance by olfactory cues were verified for bees of the studied colony. By moving whole nests or exchanging the removable entrance sections, the ability of females to distinguish their own nest from foreign ones was verified. However, as the results were identical to hitherto published studies on *O. cornuta* [5,8,10,16], the data were not shown here. In these experiments, we also checked that females accept nest tubes with a filter paper sleeve introduced into the entrance for nesting and continue to provision their nest after a short period of irritation when the insert was replaced with a fresh one.

The acceptance of filter paper inserts allowed us to analyze the composition and stability of individual scent marks of one and the same bee over the time-course of the construction of an individual nest. As most probable sources of volatiles used for marking, we analyzed volatile substances from the Dufour’s gland, cuticular hydrocarbons, and secrets from head glands (e.g., mandibular glands). To get a measure of the CHC profile as well as possible substances originating from mandibular glands, we extracted the head of the nesting bee. CHC profiles from the head were found to be similar to the residual body [18].

## 2. Materials and Methods

### 2.1. Nest Material and Sampling

Artificial nest tubes were made from hollow stems of Japanese knotweed (*Fallopia japonica*, Houttuyn). The stems were cut into segments close to raised nodes to produce naturally sealed tubes. About 200 knotweed stalks (8–12 mm internal diameter and 140–160 mm distance between node diaphragm and opening) were boxed in a wooden shelter (internal dimension 130 × 240 × 180 mm, L × W × H) as a regular nesting site for *O. cornuta* females and source of females to nest in the test tubes. The shelter was hung up under a roof facing to the southeast in a household garden in a suburban area of Berlin, Germany (52°24′28.9″ N, 13°17′46.8″ E). Beside the box, a wooden board was mounted to expose the test tubes for behavioral experiments and for collecting volatiles. Knotweed internodes with a 10 mm inner diameter were selected for experiments and trimmed to a length of about 150 mm. To collect volatiles used for nest-marking tubes, the most round, smooth internodes were chosen and carefully deburred to guarantee a close fit of the introduced filter paper sleeve made by rolling a stripe of 20 × 40 mm (Schleicher & Schnell filter circles, 125 mm). The prepared tubes were individually coded to assign tubes and nesting bees. Tubes were checked several times a day to detect the start of provisioning by a bee. The filter paper sleeve was replaced by a fresh one when the bee had filled nearly half of the tube. When the brood cells nearly reached the sleeve, the female bee was caught after depositing provisions by putting a clean glass vial over the entrance, and the second sleeve was also collected. As a control, a prepared tube was sealed by a stainless-steel wire mesh to prevent bees from nesting but exposing the sleeves to ambient conditions. Sleeves were cut into small pieces by clean scissors and extracted into a 1.5 mL glass vial by 300 µL of pure n-hexane (AppliChem). As a negative control, unused sleeves were similarly treated and extracted. Bees were killed by shortly freezing them, then they were decapitated, and the Dufour’s gland dissected. The entire head and the Dufour’s gland were extracted in separate vials by 100 and 50 µL hexane, respectively. All samples were stored at −18 °C. Heads were removed after 1 week of leaching mandibular gland substances to guarantee a stable substance mixture. In total, 25 complete sample sets consisting of two entrance sleeves, the bee head, and Dufour’s gland of the nesting bee, as well as three control entrance sleeves, were collected.

### 2.2. Analysis of Volatiles

Hexane extracts were analyzed by GC/EI-MS (Varian Saturn 2100T, Workstation Version 6.20) coupled to a CP 3900 GC (Varian) equipped with a Zebron ZB5-ms 30 m × 0.25 mm × 0.25 µm; splitless injection @ 220 °C, helium @ 1.0 mL/min, 1 min @ 100 °C, 20 °C/min to 200 °C, 5 °C/min to 300 °C, 5 min @ 300 °C; MS: EI auto @ 70 eV, mass range 40–650 m/z. Compounds were initially identified by NIST Mass Spectral Library search (Version 2.0a, build 2002) validated by retention times (Kovats indices). Mass spectra were verified by pure reference substances and reference GC/MS chromatograms of previously identified CHC compounds of the species [18]. Quantification of substances was based on the largest diagnostic mass fragment. All compounds present in quantifiable amounts were initially included in the evaluation. However, in regard to the statistical analysis, only those substances that were found to originate from the bee or environment and were present in all samples were retained, checking carefully for the occurrence of individual-specific private substances.

### 2.3. Origin of Volatiles

Only a part of the substances found in the nest entrances were also identified in the Dufour’s gland extracts. However, some of these substances might also originate substantially from the body surface. To select the substances that quantitatively originate from the Dufour’s gland, the means of the normalized peak areas for all substances were calculated for marks and gland extracts, respectively. The quotients of the peak areas of marks and glands were plotted against the substance’s Kovats indices as an estimate of volatility. Most substances were scattered around a slowly ascending regression line. All substances deserting from this band have been presumably enriched in the marks by other volatile sources and were eliminated from analysis of Dufour’s gland-related tags. However, to also consider a possible contribution of the body surface or glands of the head to the nest marks, an individual-specific total bee bouquet referring to the substances found in the paper samples was estimated. In the first step, all substances present in the head and/or Dufour’s gland extracts, as well as in nest marks, were chosen and normalized to the same peak area sum. In a second step, the means of the normalized peak areas per substance were calculated for each odor source over all samples. Then the sums of the substance means of the head and gland extracts were correlated to the mean peak area of the substance found in the nest marks. All substances were discarded if they were influenced by external volatile sources that were more than two-fold enriched in nest marks compared to the sum of the bee extracts. For all the remaining substances, the relation of peak areas was calculated. There was a clear sharp peak of ratios between one and two and a long flat tail towards higher values up to 55. Thus, for factors between one and two, a proportional contribution of both bee volatile sources to the scent mark was assumed, while for higher ratios, only the body part with the higher amount was considered as a source of this substance. In the final step, the found contribution factors were applied to the normalized peak values of both extracts of each individual bee. The estimated contribution of each source (head and gland) was then summed per substance to calculate the estimated individual bouquet per bee.

### 2.4. Statistical Analysis

Volatile bouquets were treated as self-contained compositions as no individual compounds were detectable. Due to the compositional nature of the data, the analysis was based on relative proportions. To handle a possible non-linear intensity bias due to, e.g., the body size depending on the total substance amounts, data were quantile-normalized [19] by the package ‘preprocessCore’ [20] for R [21]. Due to the compositional nature of the data, the normalized peak areas of a sample were centered log-ratio transformed [22]:$$clr(Y_{ij})=\ln\left(\frac{Y_{ij}}{g\left(Y_{j}\right)}\right)$$
where *Y_ij_* is the area of the peak *i* for bee *j,* and *g(Y_j_)* is the geometric mean of all peak areas for bee *j*. The multitude of compounds was reduced by principal component analysis (PCA, regression method), extracting all factors with an eigenvalue >1 to handle potential problems of collinearity. A hierarchical cluster analysis was employed to calculate a (dis)similarity matrix based on squared Euclidian distances (SED). Thus, small SED values indicate high similarity and vice versa. SED values were prune of extreme isolated values for further statistical analysis. A binomial-logit generalized linear model (GLM) was used to test for differences in similarity values between the samples from one and the same or different individuals. SPSS (IBM, V. 24) was deployed for all statistical analyses. The error level was set to α = 0.05. The median and range were given for samples with inhomogeneous variance. Heat maps of dissimilarities between tags and volatile sources were produced using the R pheatmap package [23] with unweighted pair group method with arithmetic mean (UPGMA) hierarchical row and column clustering, using the normalized and centered log-ratio transformed peak areas of samples as the distance metric. Columns and rows were sorted by the R dendsort package [24].

## 3. Results

### 3.1. Marking Behavior

Females of *O. cornuta* treated the inner surface of a nest tube before they started to construct brood cells carefully. They scraped the inner tube walls with their mandibles, dabbed at them with the tip of abdomen, and rubbed off the body surface on the tube walls when spiraling in the tube to work on the whole inner surface, including the introduced filter paper sleeves. This behavior was continued throughout the whole construction and provision period. Each time before leaving the nest, females treated the section of the nest tube that has not filled with brood cells yet more or less intensively.

### 3.2. Chemical Composition

Out of the 75 substances that were initially quantified and identified in extracts of the nest entrances, 59 substances could be included in the analysis (Appendix A). In general, all hexane extracts were dominated by alkanes and mono-alkenes of a preferably odd-numbered chain length. Additionally, methyl esters (ME) and ethyl esters (EE) of alkane- and alkene-acids of predominantly 16 and 18 C-atoms were found. Moreover, isopropyl ester (IPE), unsaturated carbon acids, primary alcohols, and alkanes with methyl branches were detected. The amounts of the several substances were highly variable and differed by five magnitudes based on their peak area. Most of the substances found in the nest entrances were also identified in bee samples. However, some primary aldehydes of alkanes were found in extracts of the nest entrances only. No private substances of individual bees were found.

Nest entrances comprised of a mix of a series of n-alkanes (44.7 ± 4.96% of total peak area) in the range of 21 to 31 carbon chain lengths and their corresponding alkenes (39.8 ± 9.02%). The double bonds of n-alkenes were located in the positions 5- (1.0 ± 0.33%), 7- (33.5 ± 7.58%) and 9- (5.3 ± 1.58%). Odd-numbered alkanes (42.7 ± 4.61%) and alkenes (38.7 ± 8.77%) predominated in general, with pentacosane being the prominent alkane and 7-pentacosene the dominant alkene, respectively (Appendix A; Appendix A). Carboxyl esters were found throughout the chromatogram in trace amounts (total 0.5 ± 0.29%). Aldehydes of chain lengths from 15–24 were found in variable but small amounts (total 9.1 ± 4.76%).

The Dufour’s gland extracts of nesting *O. cornuta* females were dominated by alkenes (78.7 ± 5.90%). Alkanes contributed by 19.1 ± 5.61%. In gland extracts, several esters were found at a clearly increased amount (0.5 ± 0.30%) compared to head extracts (Appendix A) and also some primary alcohols (1.3 ± 0.55%). Aldehydes were found in trace and variable amounts (0.1 ± 0.17%). In head extracts, the content of alkanes was higher (37.1 ± 3.67%), while alkenes contributed by 60.0 ± 4.41% to the CHC bouquet. The proportion of alkenes with a double bond in position 7 was increased (51.9 ± 4.27%), while the other alkenes were found in comparable proportions (5: 1.8 ± 0.34%; 9: 6.4 ± 1.31%) compared to the nest marks. Most of the esters found in the Dufour’s gland were not present or in trace amounts only (<0.1%). Aldehydes were also found in trace amounts only (0.1 ± 0.05%). The proportions of substances differed in the Dufour’s gland and in head extracts both in their absolute value as well as in their variability, although a general positive correlation was present (Appendix A).

### 3.3. Discriminability of Nest Marks

The PCA of 59 hydrocarbons found in nest marks extracted 11 factors explaining 84.89% of the variability. The SED values of nest marks were highly variable and overlapped broadly between marks from the same or different nests (Figure 1a). Marks of one and the same nest were more similar, as indicated by a median SED = 8.78 (1.78, 27.58; *n* = 25), than marks from different nests with a higher median SED = 17.64 (1.34, 53.65; *n* = 1200) based on the included substances (binominal-logit GLM: *n* = 1225, W = 17.083, *p* < 0.001). The heat map of normalized and transformed substance amounts of nest marks (Figure 2) demonstrates, on one hand, a typical tag bouquet composition pattern, as individual substances are present in all tags in comparable proportions, as indicated by the clear vertical stripes pattern. The most present and most homogenous substances are alkanes and alkenes that are typical for the CHC-pattern of the species (reddish and orange stripes). The minor compounds of the tag bouquet were more variable (bluish stripes) and contain most of the esters. On the other hand, the substance composition is variable enough to clearly distinguish tags from each other as each column (nest mark) is conspicuously different from its neighbors. However, the tags of the one and the same nest are rarely clustered together in one basic branch (clasped column captions).

### 3.4. Similarity of Bees and Marks

From 54 substances quantified in the Dufour’s gland, only 34 were also found in reliable proportions in nest marks. Due to the reduced number of 34 substances, the PCA extracted 9 factors explaining 77.01% of the total variability. Despite this restriction, the SED of Dufour’s gland extracts were right-skewed distributed and potentially highly variable, as indicated by the maximum values (Figure 1b). Nest marks were more similar to gland secretions (*n* = 34 substances) of the nesting bee compared to other bees (binominal-logit GLM, *n* = 1249, W = 4.525, *p* = 0.033; Figure 1b). Additionally, when only substances originating from Dufour’s gland were considered, the similarity of nest marks originating from one and the same nests differed from foreign nests (binominal-logit GLM, *n* = 1223, W = 12.276, *p* < 0.001; Figure 1b). A heat map based on the 34 selected substances (Figure 3) reproduced essentially the same band-pattern found in nest marks. Again, the marks of the one and the same bee rarely clustered next to each other or to the gland secretions of the respective bee. However, marks and gland secretions were also individually distinguishable with the reduced spectrum of substances.

When CHC compounds were also considered, a total of 49 substances could be factored in to estimate a hypothetic total bee bouquet. A PCA extracted 11 factors explaining 77.83% of the variability. Nest marks were again more similar to the estimated bouquet of the nesting bee compared to foreign bees (binominal-logit GLM, *n* = 1249, W = 15.540, *p* < 0.001; Figure 1c), and marks remained different when only substances originating from bees were considered (binominal-logit GLM, *n* = 1225, W = 14.993, *p* < 0.001; Figure 1c). In the case of the estimated bee bouquet, the heat map (Figure A1) only showed three clear blocks in rows, which were more dominated by the CHC compounds with a high degree of similarity (red and reddish substances in top rows). Moreover, most of the bees clustered together and were not scattered between nest marks. Again, only a few marks of the one and the same bee were combined in neighboring branches. However, all marks and bees (columns) stayed distinguishable, and no two samples were identical.

## 4. Discussion

Navigating back to and identifying their own nest are basic skills for nest-provisioning females. Confounding nests would lead to a waste of maternal investment efforts. Therefore, navigation abilities of solitary bees are in no way inferior to social species, such as honey or bumble bees. In contrast, the short-distance visual orientation has to be even better as, in dense nest aggregations, entrances may be located less than a bee body length away. Thus, it is not surprising that nest-constructing bees also use individual olfactory cues to recognize their own nest. *O. cornuta* females mark their nests too [5,16]. The composition of the nest marks was dominated by hydrocarbons and corresponded well with the species-specific CHC bouquet of nesting females [18] and resembled the bouquet of other mason bees, such as *O. lignaria* or *O. bicornis* [13,17,18]. *Osmia cornuta* females did not use any private substances for marking. This was not to be expected as the biochemistry as well as the inventory of olfactory receptors set up clear confines. Furthermore, we did not find substances different from the typical CHC bouquet of *O. cornuta* in nest labels nor in head extracts. In contrast to suggestions based on observations [10,16], we cannot support the hypothesis of using mandibular gland secretions for marking, because we did not detect any volatile substances that could be dedicated from this origin. Bees probably scratch the inner tube walls simply to clean the cavity from debris.

Marks were composed mainly by substances of Dufour’s glands secretions. As the substance composition found in the labels differs from the gland secretion, CHC substances also rubbed off the body surface and contributed to the nest tags. This basic bee-derived bouquet was enriched by substances originating from the environment. These exogenous substances arise from the nest tube itself or were brought in from flower visits. Consequently, the resulting blended marks were not identical to the (estimated) bouquet of the nesting bee, albeit the statistical similarity was higher between a bee and its own tags compared to foreign tags. Even if the assortment of substances incorporated into the statistical analysis was constrained to Dufour’s gland-derived substances, a strict mapping of bees and marks was not achieved.

Besides a low conformity of bees and tags, the differences of marks that were produced sequentially in the one and the same nest were partially large. The observed alterations document a high plasticity in the marks itself. Marks are not a static sign produced individually by every single female bee. On the contrary, the more or less individual mixture of Dufour’s gland and CHC substances are mixed with other compounds to a label that has to be learned dynamically by the nesting bee. At each visit, bees “update” their internal template to be able to recognize their nest when they return from a collecting flight despite a shifting tag. The olfactory identification is only a backup system, as *O. cornuta* females approach their nest tube by visual cues [5]. A limited variability of nest labels and a certain similarity in the labels of different bees are not a problem as long as the diversity is high enough that neighboring tubes remain distinguishable. Although the olfactory sensory space of mason bees is not known and the characterization of labels was based simply on the composition of substances detectable by hexane extraction and GC/MS analysis, the accessible variability was high enough to discriminate each and every label in our study. Even if not all substances were exploited by the bees, the marks of adjacent nests should remain distinguishable.

Substances from Dufour’s gland secretions used to identify the own nest might originally serve other purposes, e.g., antimicrobial protection [25,26]. Females were observed to impregnate the whole nesting tube and not just the entrance or vestibular area [13]. In ground-nesting bees, Dufour’s gland secretions are regularly used to line the brood cells [9,26,27]. As the composition of the gland secretions is variable to a certain degree, the lining may obtain a secondary function as part of the tag to recognize the own nest.

Labels also serve as kind of an “Occupied!” sign to announce to females searching for an unexploited cavity for nest construction that this particular tube is already in use by another female that will fight for the property of the nesting space. Due to the species-specific mixture based on the CHC pattern, an inspecting female may even access the species affiliation of the owner. The syntopic mason bees *O. cornuta* and *O. bicornis* use similar cavities and have an overlapping flight period. The CHC bouquets are composed of the same substances but show a different pattern [18]. A searching female might detect that the tube is used by an individual of another species and estimate the chances of a usurpation. Usually, *O. cornuta* females are able to take over nests from females of the smaller species, *O. bicornis* (personal observations). Whether a searching female is also able to assess the constitution of an intraspecific nest owner and to estimate her chances to usurp the cavity remains to be investigated. Observations of an intensified nest marking behavior by the resident bee at the entrance in response to intrusion by another female in *O. lignaria* [13] and *M. rotundata* [12] point in this direction.

## Figures and Tables

**Figure 1 insects-12-00843-f001:**
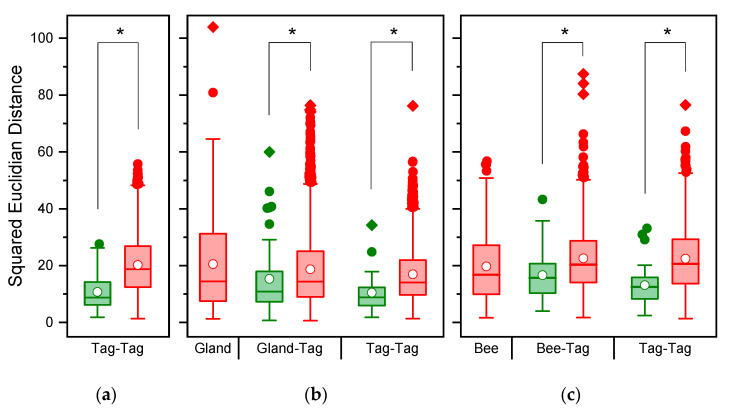
(Dis)-Similarity of chemical composition between scent tags of nests and origins of chemical compounds. The similarity is given as squared Euclidian distances, therefore, smaller values indicate a higher similarity. (**a**) Comparison of nest tags including all 59 substances; (**b**) Comparison of Dufour’s gland extracts (34 compounds) and nest tags. (**c**) Comparison of the estimated bee bouquet (49 compounds) and nest tags. Green: samples of one and the same bee, red: samples of different bees; boxes: 25th, 50th, and 75th percentiles, white dots: the mean; whiskers: the range of 1.5 IQR, colored dots: outliers, squares: extremes, * *p* < 0.05.

**Figure 2 insects-12-00843-f002:**
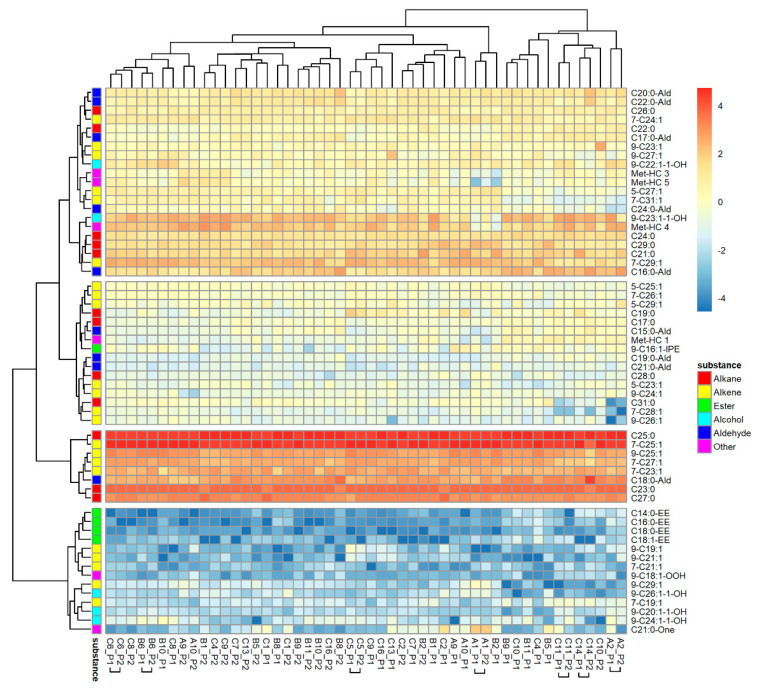
Composition of individual scent marks of *O. cornuta* nests. Compounds are arranged in rows, and odor samples are in columns of the heat map. Samples are specified by the ID of the nest and the number of the filter paper sleeve. Samples of one nest that cluster together are marked by horizontal squared brackets.

**Figure 3 insects-12-00843-f003:**
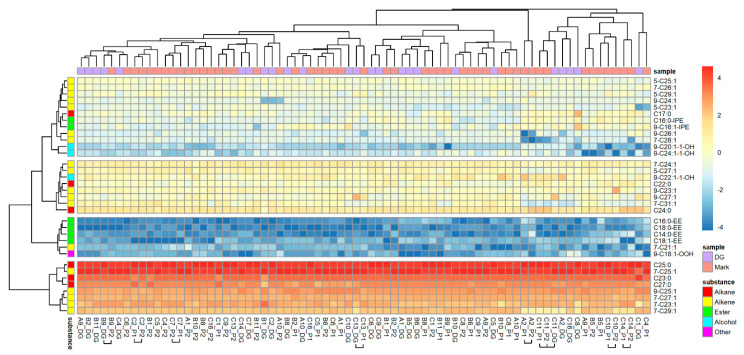
Composition of Dufour’s gland extracts and individual scent marks of *O. cornuta* nests (34 substances). Compounds are arranged in rows, and odor samples are in columns of the heat map. Odor samples are specified by the ID of the nest and origin of the sample (gland of resident bee/number of the filter paper sleeve). Samples of one nest that cluster together are marked by horizontal squared brackets. Origin of samples and substance classes are color-coded for a quick overview.

## Data Availability

The dataset generated and analyzed during the current study is available from Share_it—Open Access und Forschungsdaten-Repositorium der Hochschulbibliotheken in Sachsen-Anhalt at http://dx.doi.org/10.25673/38027 (accessed on 15 May 2021).

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
