# Peer review of "Individual Scent-Marks of Nest Entrances in the Solitary Bee, Osmia cornuta (Hymenoptera: Apoidea)"

_insects, 2021, doi:10.3390/insects12090843_

Round 1

Reviewer 1 Report

The article is interesting, because it demonstrates that insects could be clever.  The authors have made the hard experimental work. Methods are described in detail. The results are novel and well-documented. The main conclusion is that the bee always remembers the scent of its own nest in spite of this scent is changing over the time. It means that the bee should always learn new scent of its nest. The manuscript is well written.

Specific comments:

Line 40 It would be useful to indicate, how long does the bee usually live in the nest (two weeks?) and to estimate, how many times it has to return to its nest.

Line 51 Please, add the author to the species mentioned for the first time: Megachile rotundata.

Line 55 Please, indicate, where Dufour’s gland is situated.

Line 58 Please, add the author to the species mentioned for the first time: Osmia lignaria

Line 66 Please, add the author to the species mentioned for the first time: Osmia cornuta

Line 72  Were the cited studies carried out on Osmia cornuta or on other species? If other species was previously studied, your results are not identical to the previously published. Therefore, I would recommend the description of these results into your manuscript. All experimental  results are valuable, even those which just confirm previous studies.

Lines 72-79 These detailed observations on the presumable marking behavior are interesting and should be probably moved from the “Introduction” to the “Results” section. Probably only the short description is appropriate in the introduction.

Line 97 Since the manuscript is in English, please correct  “52°24'28.9"N, 13°17'46.8"O” to “52°24'28.9"N, 13°17'46.8"E”

Line 182 And what about other 16 substrates?

Line 282 “O. cornuta” – Latin names in the beginning of the sentence should be written in full -  Osmia cornuta.

Line 288 Bess - ?

Author Response

Dear reviewer,

We are grateful for your vulnerable advices and followed your suggestions without exception. We feel that the points are important and we readily heeded your advices. Please find our detailed answers to your questions in the attachment.

Reviewer 2 Report

It’s an interesting manuscript describing a significant experimental work that produced a great amount of data. The authors have made a significant effort to present the complexity of the chemical analyses in a synthetic way and by graphic representations. However, the description of the methodology for the analysis of the different volatilomes was not easy to read. I found some sentences unclear, a few typos, and missing words. I had difficulties to perceive what resorts to two different points: the origin the compounds and their value as an individual marker. I recommend a revision of this section and the addition of a graphical abstract of the procedures also showing the names of the different reduced volatile organic compound (VOC)-lists.

Line 114 “Heads were removed after 1 week of leaching mandibular gland substances to guarantee a stable substance mixture”. I understood (line 113) that mandibular glands were removed by dissection and extracted in different vials than the rest of the head so what is a “stable substance mixture”: make sure to extract all components of mandibular glands by a longer maceration? Isolate compounds from gland and head by removing the head (which seems contradictory with the preceding sentence)?

Line 137: “All substances found in the nest entrance to be enriched compared to the glands and were substantially present also in head samples were eliminated from analysis of Dufour’s gland related tags”.  Should we read “that were substantially present”? “substantially” is not precise enough. I understood from this sentence that you based the attribution of an origin to nest VOCs on the comparison of their proportions between the postulated source (gland) and the nest (filter paper). However, the studied VOCs have very different volatilities and quite quickly their proportions will be altered after the deposition compared to the original proportions. Therefore, the relative proportion of a low volatility product will be higher in the nest after a few seconds than in the gland which produced it (apparent enrichment). This does not necessarily imply that the "enriched" product is of extra glandular origin.

Line 139 “However, nest marking substances could originate also from the body surface or glands of the head” This sentence repeats more or less line 134

Lines 140- 141 “To estimate a mean bouquet of the bee referring to the substance amounts fond in the paper samples” found instead of fond ? But at this point, I get confused about what part of the data are presented for each heat map. Please make a clear correspondence with figures.

Line 158 “Volatiles were treated as bouquets as no individual compounds were detectable” Not sure to understand what do you mean, because the peaks are attributed to individual chemicals so it seems to me that individual compounds are detected and measured.

Line 288 “Bess scratch the inner tube” Bees instead of bess

You compared complete volatilomes and showed that they are sufficiently different to be discriminable, but also too variable for the discrimination to be reliable. This conclusion assumes that bees use the full panel of VOCs. Indeed, insects in general face to very complex mixtures (a plant volatilome for instance contains more than one hundred components), but a phytophagous insect makes its choice basing on a fraction only of the plant odorants. Depending on filtering by their ORs (not all VOCs are detected) and integration by the olfactory centers (not all VOCs have a positive valence), only parts of the whole nest-volatilomes may be used by the bee. On the contrary, the overall analysis may provide a blurred picture, because biased in favor of compounds that are very variable and in large amounts, without any correlation of their real importance to bee olfaction, and the real “signature” might rely on a few components very well detected and perceived (salient)?

Finally, I have two comments. The nest-odor can also evolve as the bee introduces new food material from different plants and I like the idea of the necessity for plasticity in olfactory identification.

It is often read that the grouping of artificial nests for “solitary” bees is problematic because it is unnatural and leads to competition between females and sensibility to parasites and diseases. However, in the introduction you indicate that the potential nesting sites being often localized, the nests are often naturally very close to each other and your experiments show that bees can deal with nest closeness. Do you think your observations could contribute, for instance by setting an “optimal” density for artificial nests, to conservation of pollinator populations?

Author Response

(The authors gave the same response as above.)

Reviewer 3 Report

Authors are studying the profiles of individual scent-marks of the solitary bee Osmia cornuta nests. The composition looks like the complex mix originates from Dufour’s gland, cuticle and external volatiles. The paper is important and well done, with very interesting note/observation in the Discussion regarding the sequestering of nests by O.cornuta from other species.

Methodological questions:

  1. As all chemicals are collected by the filter paper, is it possible the refuse by the solitary bee to touch the laboratory filter paper as biologically external material?
  2. Could some important substances be escaped from hexane extraction?

Minor points for the text:

  1. Figure S1-S3 are now in the supplemental materials, they could stay well also in the main text.
  2. Line 171, the abbreviation GLM and UPGMAplease explain at first mentioning in the text.
  3. Figure 1 legend, line 233, reference to panel c) must be in bold character.

The question to authors, slightly out of the main topic of the paper: which is the precision of nest recognition by Osmia cornuta? Approximately, which is the percent of error in the recognition of own nest?

Author Response

(The authors gave the same response as above.)

Round 2

Reviewer 2 Report

The revision has made the paper easier to read. Not sure to understand the acknowledgement sentence "The authors wish to thank the referees for their vulnerable comments"